# Vaccination against COVID-19: Factors That Influence Vaccine Hesitancy among an Ethnically Diverse Community in the UK

**DOI:** 10.3390/vaccines10010106

**Published:** 2022-01-11

**Authors:** Erica Jane Cook, Elizabeth Elliott, Alfredo Gaitan, Ifunanya Nduka, Sally Cartwright, Chimeme Egbutah, Gurch Randhawa, Muhammad Waqar, Nasreen Ali

**Affiliations:** 1School of Psychology, University of Bedfordshire, Luton LU1 3JU, UK; alfredo.gaitan@beds.ac.uk; 2Public Health, Luton Borough Council, Arndale House, 37 the Mall, Luton LU1 2LJ, UK; Elizabeth.Elliott@luton.gov.uk (E.E.); Sally.Cartwright@luton.gov.uk (S.C.); Chimeme.Egbutah@luton.gov.uk (C.E.); 3Institute for Health Research, University of Bedfordshire, Putteridge Bury Campus, Hitchin Road, Luton LU2 8LE, UK; Ifunanya.Nduka@beds.ac.uk (I.N.); gurch.randhawa@beds.ac.uk (G.R.); Muhammad.waqar@beds.ac.uk (M.W.); nasreen.ali@beds.ac.uk (N.A.)

**Keywords:** COVID-19, vaccine hesitancy, vaccine uptake, ethnicity, inequalities, health beliefs

## Abstract

The UK’s minority ethnic population, despite being at higher risk of COVID-19 and experiencing poorer health outcomes, continue to have lower uptake of the COVID-19 vaccine compared with their white British counterparts. Given the importance of the vaccination programme in improving health outcomes, this research sought to examine the influential factors that impact the decision to accept the COVID-19 vaccination among an ethnically diverse community. A total of 1058 residents from Luton, UK, a large town with an ethnically diverse population, completed a community survey. Questions centred around uptake or individuals’ intentions to accept the offer of COVID-19 vaccination alongside demographics, knowledge, and views on the vaccine. A binary logistic regression analysis was conducted to determine the most significant predictors of vaccine hesitancy, while respondents’ reasons for not getting vaccinated were identified using qualitative content analysis. Findings revealed that age and ethnicity were the only sociodemographic factors to predict vaccine hesitancy. Knowledge of symptoms and transmission routes, alongside ensuring information about COVID-19 was objectively sourced, were all identified as protective factors against vaccine hesitancy. Qualitative analysis revealed that ‘lack of trust in government/authorities’ and ‘concern of the speed of vaccine development’ were the most common reasons for non-uptake. This research reinforces the importance of age, ethnicity, and knowledge as influential factors in predicting vaccine hesitancy. Further, this study uncovers some of the barriers of uptake that can be utilised in developing promotional campaigns to reduce vaccine hesitancy in certain sections of the diverse UK population.

## 1. Introduction

The recent Public Health England (PHE) report on disparities in risk and outcomes from COVID-19 identified that men from black ethnic groups were over three times more likely to die from COVID-19 than people of white ethnic groups, even when accounting for age, sex, deprivation, and region [1]. People from Chinese, Indian, Pakistani, other Asian, Caribbean, and other black ethnic backgrounds had between 10% and 50% higher risk of death when compared with the white British population [2]. Worse outcomes from COVID-19 have also been exacerbated by sociodemographic factors associated with poverty and deprivation, including overcrowded housing, income inequality, occupations with a higher risk of COVID-19 exposure, need to use public transportation, and poorer health experiences of healthcare alongside historic racism [1].

The successful development and implementation of vaccines are currently our most effective defence against SARS-CoV2 infection. The vaccines BNT162b2 and ChAdOx1-S are the most commonly used in the UK, shown to be 90% [3] and 82% [4] effective, respectively, against severe disease to the original variant. More recently, both vaccines have been shown to deliver robust protection against severe disease, hospitalisation, and mortality from the Alpha and Delta variants of the coronavirus present in the UK [5]. However, the success of a vaccination programme to provide ‘herd immunity’ (the proportion of subjects with immunity in each population) remains dependent on a high proportion of the population being vaccinated [6]. The exact number is contingent on a range of factors, including the transmissibility of variants, the efficacy of vaccines, and population behaviour. However, at least 70% of the population need to be vaccinated to provide sufficient protection against the currently circulating Delta variant to see a reduction in rates [7]. From the start of the vaccination programme, it became increasingly clear that the introduction of the COVID-19 vaccination programme had the potential to exacerbate some of these existing inequalities [8].

In the UK, the COVID-19 vaccination programme began early December 2020. It was rolled out to the whole population in ‘cohorts’ with the highest risk eligible for vaccination earliest and subsequent cohorts invited as supplies allowed. Cohorts were identified by the Joint Committee on Vaccinations and Immunisations (JCVI) and included an assessment of risk that included age, working in health or social care, and existing medical conditions. By July 2021, the offer of the vaccine was made available to all adults aged 18 and over. The overall uptake of the vaccine in the UK has been high, with over 85% of all eligible people (18 years and over) taking up the first vaccine by mid-July 2021 [9]. However, uptake has been disproportionately lower among the UK’s ethnically diverse communities, with the lowest rates found among those from a Pakistani (78.4%), black Caribbean, and black African (66.8% and 71.2%, respectively) ethnic background [10].

Vaccine hesitancy, characterised as ‘uncertainty and ambivalence about vaccination’ [11] is markedly higher across black and south Asian communities [2,12,13,14,15] compared with the white British population. International evidence also suggests that this issue is not unique to the UK, whereby minority ethnic groups in the United States [15,16,17,18], Qatar [19], and Israel [20] have all demonstrated higher levels of vaccine hesitancy when compared with their majority populations. Current evidence suggests that since the start of the UK vaccine programme rollout, identified rates of vaccine hesitancy within the population are around 18% [13]. Whilst, hesitancy has been shown to reduce over time [12], with only 4% of the population being revealed as vaccine-hesitant, this remains higher in Muslim (14%) and black British populations (21%) [21].

Sociodemographic factors that include being female [13], younger (aged 16–24 years) [12,13,16], having lower educational attainment (no formal qualifications) [13,14,20], and lower income (<GBP 16,000) [12,14,17] have all been shown to impact vaccine hesitancy. Research taking place before and during the COVID-19 vaccine programme rollout has identified a range of barriers that are shown to influence an individual’s view of the vaccine and whether they would be considered vaccine-hesitant. The most common reasons cited involve access barriers, which include disparities in vaccine coverage [11,22], lack of trust in statutory authorities [2,15,17,23], and concerns about the vaccine including unknown future effects [2,24] alongside concerns regarding the efficacy [12,16], potential side effects [2,12,17,23], and speed of vaccine development [12,16]. Perceptions that COVID-19 is not serious [17,19], lack of perceived need [23], and lack of awareness of entitlement to vaccine [23] have also been shown to be significant obstacles to vaccination.

Uptake of the COVID-19 vaccine has been disproportionately lower among the UK’s minority ethnic population [9], with vaccine hesitancy shown to be highest among black, Bangladeshi, and Pakistani populations [13,25]. However, if we are going to address this issue in a bid to reduce inequalities and co-develop solutions, we must understand the influential factors across a representative sample to ensure views are reflective of their respective communities. There is increasing evidence that has highlighted the interrelated social determinants that influence the intention to have the COVID-19 vaccination among ethnically diverse communities in the UK [13,14]. However, there is less known about how these determinants have changed since the rollout of the COVID-19 vaccination programme.

## 2. Materials and Methods

This study used an online and paper-based survey disseminated between January 2021 and March 2021 as part of the wider community engagement programme Talk Listen Change (TLC), funded by Luton Borough Council. The overall aim of this project was to engage with local communities to discover their views about the disproportionate impact of COVID-19, tackling health inequalities, and co-developing solutions. This paper reports the findings from the community survey, which investigated individuals’ intentions to take up the COVID-19 vaccination programme. It will add to the understanding of these communities and further the current evidence base around the causes of vaccine hesitancy to support measures to increase vaccine take up in these communities.

### 2.1. Participants and Procedure

Luton is an ethnically diverse town situated in the East of England with a population of around 216,000 people. Luton is only one of three towns in the UK where less than 50% of the population identify as white British [26]. Luton also experiences high levels of deprivation (as measured by the Index Multiple Deprivation [27]) compared with other parts of the UK [28], where large sections of the ethnically diverse community reside.

The inclusion criteria for the questionnaire required participants to be 16 years of age and older, living in Luton, UK, and identifying as a member of the Pakistani, Bangladeshi, Indian, black African, or Caribbean community. A systematic multi-pronged approach was taken to recruit participants to achieve a representative sample of the diverse population. First, an online survey was put onto Qualtrics (Qualtrics, Seattle, WA, USA) and disseminated through various online channels and platforms, including the local council website, Facebook (Meta Platforms Inc., Menlo Park, CA, USA), Instagram (Meta Platforms Inc., Menlo Park, CA, USA), WhatsApp (Meta Platforms Inc., Menlo Park, CA, USA) groups, and Twitter (Twitter Inc., San Francisco, CA, USA) pages, incorporating hashtags, photos, and links to boost engagement. Posters containing the TLC branding with a QR code link to the questionnaire were also printed and disseminated across Luton in community shops and other venues. In addition, engagement with several local community radio stations (Inspire FM, Pangtengo FM), including hourly adverts in both English and Urdu, also directed individuals to the questionnaire via a university website page which hosted lay information for the local community as well as links to take part in the study. Surveys were also printed as brochures and disseminated through community groups, places of worship, and food banks to increase recruitment. COVID-19 Champions, TLC Community Researchers, and local councillors were also recruited for this study and who distributed the printed survey through their networks using snowballing techniques to engage with hard-to-reach individuals. TLC Community Researchers interviewer-administered the survey for those people who were unable to read the questionnaire. A total of 1058 individuals completed the survey.

### 2.2. Questionnaire

#### 2.2.1. Sociodemographic Characteristics and Health

Participants provided self-reported sociodemographic data for age, gender (female, male, transgender, non-binary, other), self-ascribed ethnicity (Indian, Pakistani, Bangladeshi, African, Caribbean, mixed ethnic background, other), religion (Christian, Buddhist, Hindu, Jewish, Muslim, Sikh, none, other), and the highest level of education (no qualifications, GCSE or equivalent, A-level or equivalent, first degree (e.g., BSc, BA), higher degree (e.g., MSc, MA), and other). Full postcode was requested and matched to the Index of Multiple Deprivation (IMD) 2015 [27] score using Geo Convert software. All IMD measures were divided into ten deprivation deciles, with each quintile comprising 10% of the population of England. A lower decile indicated increasing deprivation. Participants were asked if they currently live with a chronic disease (Yes/No) or if anyone in their household other than them currently lived with a chronic disease (e.g., chronic lung disease, diabetes, cardiovascular disease, or chronic renal or liver disease) (Yes/No).

#### 2.2.2. Knowledge of Symptoms of COVID-19

Participants were provided with a total of 22 symptoms of COVID-19, which represented both common (e.g., fever, cough, loss of taste), less common (e.g., headaches, sore throat, joint aches), and severe symptoms (e.g., difficulty breathing, altered consciousness). The symptoms were taken from a published survey [29,30], with symptoms indicated by the World Health Organization [31] and the Centers for Disease Control and Prevention [32]. Participants were asked to indicate whether they believed each was a symptom of COVID-19 (yes = 1; no/do not know = 0). Responses were added, with a higher total score indicating a higher level of knowledge of COVID-19 symptoms.

#### 2.2.3. Knowledge of Routes of Transmission

Participants were provided with three statements [29], which represented three different routes of transmission of COVID-19, and were asked to comment whether each was a transmission route (yes = 1; no/do not know = 0). The three routes included (1) close contact with an infected person who has symptoms, (2) close contact with an infected person even if they are not showing symptoms of infection, and (3) contact with surfaces an infected person has touched. Responses were added, with a higher total score indicating a higher level of knowledge of transmission routes.

#### 2.2.4. Assessing Trustworthiness of Sources

Participants were provided three statements that examined to what extent participants assessed the reliability of sources they accessed about COVID-19. Statements included (1) ‘I ensure that information I access is from a trusted, reputable, well-known source’, (2) ‘I compare information I access with other trusted sources to ensure it is accurate’, and (3) ‘I access information objectively to determine the information I read is presented in a balanced, reasonable and unbiased manner’. Responses were recorded using a 5-point Likert scale which ranged from (1) strongly disagree to (5) strongly agree.

#### 2.2.5. COVID-19 Vaccine Intention 

Participants were asked whether they had received the COVID-19 vaccination at the time of completion, with yes and no response options. If participants selected ‘no’, they were asked, ‘How likely are you to accept the COVID-19 vaccination when it becomes available?’ Participants were then provided five response options ‘I definitely will accept the vaccine’, ‘I am likely to accept the vaccine if I receive information to reassure me about its safety’, ‘I don’t feel I need the vaccine for myself, but I will accept it if it protects the wider community from COVID-19′, ‘I will only accept the vaccine if it is a requirement for travelling abroad or going to work’, and ‘I definitely will not accept the vaccine under any circumstances’.

Participants who selected the response option ‘I definitely will not accept the vaccine under any circumstances’ were then directed to two open-text questions: (Q1) ‘Why have you decided not to accept the COVID-19 vaccine when it is available?’ to gain insight for the reasons why individuals chose not to get vaccinated, and (Q2) ‘What, if anything would persuade you to accept the COVID-19 vaccine when it is available?’ to ascertain what if anything might change respondents’ minds in this respect.

### 2.3. Statistical Analysis

To assess vaccine hesitancy, a dichotomous variable was developed. Participants who had already disclosed they had the vaccination alongside those who selected ‘I definitely will accept the vaccine’ were categorised as ‘vaccine-willing’. Participants who chose the response options ‘I am likely to accept the vaccine if I receive information to reassure me about its safety’, ‘I don’t feel I need the vaccine for myself, but I will accept it if it protects the wider community from COVID-19′, and ‘I will only accept the vaccine if it is a requirement for travelling abroad or going to work’ were all categorised as ‘vaccine-hesitant’.

Categorical variables for all explanatory variables were analysed using chi-square goodness-of-fit analysis and Fisher’s exact tests. Adjusted standardised residuals (ASRs) were calculated to indicate the importance of the cell to the ultimate chi-square value, which took account of the overall sample size. Therefore, when reporting the results, the ASR values were used to indicate significance, i.e., ASR values of 3.09 (*p* < 0.001), 2.6 (*p* < 0.01), and 2 (*p* < 0.05) indicated importance, with anything below two deemed non-significant (*p* > 0.05). A binary logistic regression analysis was conducted with sociodemographic variables alongside knowledge of symptoms and transmission entered as explanatory variables. All statistical tests were completed using IBM SPSS Version 26 (IBM, New York, US); two-tailed significance was assumed at *p* < 0.05.

### 2.4. Qualitative Content Analysis

The open text responses to Q1 and Q2 were analysed using qualitative content analysis, conducted in three stages. Stage 1: Responses to each question were entered on an Excel (Microsoft, Redmond, US) spreadsheet. Statements in each response were identified since some answers contained more than one statement. A sample composed of 25% of all statements for each question was selected. A researcher (A.G.) generated codes for these statements and provided definitions to reflect the codes, which were then grouped by topic. Stage 2: A new sample, also 25% of the total, was selected. Two coders (different from the researcher) (L.D. and I.N.) coded the two samples (50% of the total) using the set of codes and definitions generated in Stage 1. Simple percentages of the agreement for each question were calculated. The researcher examined all disagreements and definitions were altered, or new codes added if necessary. Agreement for Q1 and Q2 was 69.5% and 68%, respectively. Stage 3: The remaining 50% of the statements were coded (I.N.). All the coding was completed and collated on an Excel spreadsheet. Frequencies for each code and the questions were also calculated. Verbatim statements were chosen to illustrate the most frequent codes in each topic and arranged to construct meaningful accounts of respondents’ views on each of the two questions. The coding frameworks are presented in Appendix A.

## 3. Results

A total of 1058 participants took part in the survey. Most participants completed the paper-based survey (*n* = 761; 72.7%), with the remaining participants (27.3%; *n* = 286) completing the questionnaire online. The distribution of the sociodemographic characteristics of the sample and target population (where relevant) is provided in Table 1.

Of those asked, 322 (30.4%) participants stated that they had had at least one dose of the COVID-19 vaccination at the time of completion, while 612 participants (57.8%) stated that they had not yet had the vaccination (Table 2). Responses were missing for 124 participants (11.7%). From those who selected ‘no’, 596 participants completed the question regarding the intention to have the COVID-19 vaccination.

### 3.1. Quantitative Analysis

Chi-square analysis confirmed that there were significant differences in vaccine hesitancy between age groups (*X*^2^ = 55.87, df = 4, *p* < 0.001), with participants aged 30 years and younger significantly more likely to be vaccine-hesitant (ASR 5.8) and those aged 51–65 (ASR 3.6) and aged 65+ years (ASR 4.9) vaccine-willing. There was also a significant difference found for ethnicity (*X*^2^ = 45.49, df = 8, *p* < 0.001), whereby the analysis confirmed that Indian participants were the only ethnic group significantly found to more likely to be vaccine-willing (ASR = 5.9; *p* < 0.001), with vaccine hesitancy found to be significantly higher among black African (ASR 2.3; *p* < 0.05) and black Caribbean (ASR 2.1; *p* < 0.05) participants. Additionally, a significant difference in vaccine hesitancy was found depending on whether the participant had a self-disclosed chronic disease or not (*X*^2^ = 7.43, df = 1, *p* = 0.006), whereby the results confirm that participants were more likely to be vaccine-willing if they had a chronic health condition (ASR 2.7; *p* < 0.01). No significant differences were found for gender or education level.

A binary logistic regression was also conducted on the intention to receive the vaccine, with knowledge of symptoms and transmission alongside sociodemographic variables entered as explanatory variables. This model was significant, i.e., the probability of obtaining this chi-square statistic given that the null was true (*X*^2^ = 114.27, df = 27, *p* < 0.001) (Table 3). The regression identified a significant difference for age, with increased age linked to lower levels of vaccine hesitancy, with an odds ratio of 0.95 (CI 0.93–0.97, *p* < 0.001). The findings also confirmed a significant difference found for ethnicity (when holding all other independent variables constant). Black African, Caribbean, and mixed white and black Caribbean were significantly more likely to be vaccine-hesitant than those self-identified as Indian. Levels of knowledge of both symptoms and transmission routes significantly reduced the likelihood of vaccine hesitancy, with an odds ratio of 0.94 (CI 0.89–0.99, *p* < 0.01) and 0.63 (CI 0.45–0.86, *p* < 0.001), respectfully. The results also confirm that those who assessed information objectively were less likely to be vaccine-hesitant, with an odds ratio of 0.56 (CI 0.38–0.80, *p* < 0.001); however, no significant difference was not found in relation to ensuring information comes from a trusted, reputable, well-known source and comparing information accessed with other reliable sources. Religion, self-disclosed chronic health conditions, education level, and deprivation (IMD score) made no significant contribution to explaining vaccine hesitancy. A Pearson’s correlation was also conducted and found a significant positive correlation between a higher education qualification and knowledge of transmission routes r(1042) = 0.08, *p* = 0.009, but not for the awareness of symptoms r(1042) = 0.01, *p* > 0.05.

### 3.2. Qualitative Content Analysis

#### 3.2.1. Reasons for Not Getting Vaccinated

A total of 44 responses (45 statements) from 53 participants invited (83% response rate) were obtained. Eleven codes were used to analyse these responses, grouped into four topics: beliefs (47%), health (31%), information (13%), and other (9%).

Twenty-one statements reflected beliefs. Of these, an almost equal number of statements referred to lack of trust in the vaccine (9) and lack of trust in the government (8). The first code included simple statements such as ‘I don’t trust it’ or ‘lack of trust in the vaccine’, which gave the label to the code. Other statements indicated insufficient research or concerns that the vaccine’s approval had been rushed: ‘Vaccines require years of research and do not fully protect you from transmission.’ A similar response, but more elaborate, stated, ‘It has not gone through phase III trials. It has been emergency authorised and is not an approved vaccine for COVID-19—SEE FDA + Pfizer website.’ Another respondent stated a more fundamental reason: ‘I believe the vaccine is not safe for our body.’ One respondent expressed the lack of trust in the government simply as ‘I don’t trust the government’ or ‘mistrust of the UK government’. Another wrote, ‘Too many lies lots of people die.’ Others expressed mistrust of the pharmaceutical companies: ‘I have no trust what’s in it. Companies are actually making £’ and ‘There is no trust in the product.’ The remaining two codes had too low frequencies.

The next group referred to statements that voiced a health-related reason for not getting vaccinated, although frequencies for these codes were notably low. The first type of reason directed to the side effects of the vaccine. Some responses referred to *facts*: ‘I am not comfortable because lots of vaccination people are all facing side effects.’ In contrast, others referred to information: ‘A lot of negative information and chances of side effects it could have’ or talk ‘With a lot of talk in the community and worldwide, doctors discussing in their live webinars the dangers of the vaccine and the evidence to prove it.’ Interestingly, a smaller number of responses expressed a lack of need: ‘I am perfectly in great health’ or ‘(I am) not in danger.’ A more detailed response was, ‘I don’t feel the need to take this vaccine the same way I don’t feel the need to take some of the other vaccines.’

#### 3.2.2. What Would Change Your Mind about Getting Vaccinated?

Participants who did not wish to be vaccinated were also asked what would be needed to change their minds. Nineteen participants answered the question, and 22 statements were coded in their responses. The most significant proportion related to information (45%; *n* = 10), with others referring to health (14%; *n* = 3), with seven individual suggestions grouped as ‘other’ (32%; *n* = 7) (Appendix A).

Seven of the ten statements related to information referred to the *need for evidence* to back up claims made. These included ‘Data to back up the claim’. For another respondent, this meant ‘Once it has been tested long enough as I don’t believe the stats to be true’, which is like ‘More trials of the jab’. A more detailed view was expressed in the following statement: ‘FDA approval, a view of the long-term effectiveness and effect, and scientific evidence to support these views, given the number of cases and deaths are charting similarly to nations not as advanced in their rollout.’ For two more respondents, ‘Information on long-term side effects of vaccine’ would be decisive (e.g., ‘Unless I get to know the long-term side effects of the vaccine’).

Three health-related reasons included reassurance that the vaccine does not affect fertility or pose any risks during breastfeeding, but again these were the concerns of only two individuals. Under the category of ‘other’, we grouped seven statements. Two of the participants would refer to ‘Someone from my family or high profile not experiencing side effects’, a further two ‘to having access to vaccines abroad’, and the final two ‘only if it was made compulsory’.

## 4. Discussion

This study revealed that age and ethnicity were the only sociodemographic factors to predict vaccine hesitancy, with increased age and being Indian linked to lower levels of vaccine hesitancy. Furthermore, it was identified that increased knowledge of COVID-19 symptoms and transmission routes were protective against vaccine hesitancy. This is supported by previous research, which has highlighted a correlation between lower cognition and increased vaccine hesitancy [24]. The findings also highlighted a significant relationship between knowledge of COVID-19 transmission routes and educational attainment, suggesting that those with higher educational attainment are more likely to be more aware of transmission routes. However, this was not found for knowledge of COVID-19 symptoms. This supports previous research that has also acknowledged the critical role of educational attainment on vaccine hesitancy [13,19] and reinforces the importance of ensuring that knowledge disseminated regarding COVID-19 remains tailored to suit differing levels of educational attainment. Interestingly, the findings also reveal that accessing unbiased information related to COVID-19 was linked to lower levels of vaccine hesitancy. This, therefore, highlights the importance of educating population groups on how to validate information to ensure their vaccination decision is drawn on objective, credible evidence.

Qualitative content analysis of open text comments revealed useful information regarding why some communities may be vaccine-hesitant. The most common reasons for not wanting to have a vaccine centred on lack of trust, primarily related to the wider government and the vaccine, including concerns of vaccine side effects. This finding is consistent with previous research both nationally [2,12,23] and internationally [15,16,17,18]. Trust, or lack of, is revealed as a consequence of systemic racism and discrimination [11,34], in a society where ethnic inequalities and disparities in health and healthcare continue to persist [34]. The findings highlighted that whilst knowledge surrounding the vaccines longer-term safety and efficacy were viewed as the most important factors to encourage those to have a vaccine in the future, trust in the evidence and the institutions that deliver vaccination campaigns is crucial in influencing their decision.

These findings provide support for previously published evidence and uncovered new evidence regarding the importance of knowledge about COVID-19 (including transmission routes and the wide range of symptoms of COVID-19) and validating sources as a significant determinant of lower vaccine hesitancy. This research has implications for the delivery of communication of vaccine messages. If we can improve individuals’ knowledge around COVID-19 within these communities, this can support the delivery of take-up of the vaccine. This is key, as Luton currently has the third lowest uptake of both first and second doses of the vaccine outside London in the UK [9]. The qualitative data highlighted that trust in information is the most critical issue around acceptance of messages; however, communities would be more willing to engage with vaccine-related messages by delivering information through trusted sources. Further work is now needed to identify the most helpful way to engage with these communities.

### Strengths and Limitations

This study presents a large cross-sectional study that explored the influential factors that affect the uptake of the COVID-19 vaccine among an ethnically diverse population in the UK. The inclusion of community researchers alongside a flexible approach to recruitment enabled us to achieve a large ethnically diverse sample and give a voice to traditionally underrepresented groups in health research, which have enabled us to understand better the views related to vaccine hesitancy. It is also important to note that this survey took place during the rollout of the COVID-19 vaccination programme. Over 30% of the respondents were already vaccinated when they took part, which provided a real-time snapshot of vaccine hesitancy.

Nonetheless, some limitations are noteworthy. First, whilst this study provides the views of an ethnically diverse population within the UK, it is essential to note that the investigation is not and does not contend to be nationally representative of the wider UK population. Further, as with other cross-sectional studies, the evidence presented is not causal. Therefore, it is not possible to consider how views across time may have changed, particularly in a constantly moving landscape much influenced by broader socio-political factors. There may be particular cultural and historical experiences not captured within this study which may make minority populations feel more vulnerable to vaccine hesitancy. Future research would be well placed to offer a more detailed exploration of these potential influences.

In conclusion, this study contributes to current evidence through understanding the prominent factors that may influence COVID-19 vaccine hesitancy among ethnically diverse communities. Lower levels of knowledge, both of the symptoms and routes of transmission, were significant predictors of vaccine hesitancy. Therefore, effective tailored communication and engagement may potentially reduce vaccine hesitancy in these populations. However, further work is now needed to investigate the most effective approaches to engaging and communicating with ethnically diverse communities around COVID-19 vaccines. Supporting communities to validate information accessed regarding COVID-19 to ensure it is objective may also assist in the challenge to reduce vaccine hesitancy. These findings can be utilised in developing programmes and events around reducing vaccine hesitancy in certain sections of the diverse UK population.

## Figures and Tables

**Table 1 vaccines-10-00106-t001:** Sociodemographic characteristics of the sample and target population in Luton, UK.

Sociodemographic Characteristic	Subcategory	Sample *N*	Sample (Luton) %
Age	<30	361	34.1 (20)
	31–40	200	18.9 (16.4)
	41–50	244	23.1 (12.4)
	51–65	178	16.8 (15.1)
	65+	56	5.3 (0.7)
Gender	Male	414	39.1 (51)
	Female	634	59.9 (48)
	Other	10	1.0 (-)
Ethnicity	Asian: Indian	132	12.5 (5.2)
	Asian: Pakistani	502	48.1 (14.4)
	Asian: Bangladeshi	176	16.6 (6.7)
	Black African	91	8.6 (4.5)
	Black Caribbean	75	7.1 (4.0)
	Mixed: white and black Caribbean	9	0.9 (1.9)
	Mixed ethnic background: white and black African	6	0.6 (0.5)
	Mixed ethnic background: white and Asian	21	2.0 (0.9)
	Any other ethnic background	26	2.5 (0.7)
Education Status	No formal qualifications	123	11.6
	GCSE or equivalent	178	16.8
	A-Level or equivalent	233	22.0
	First Degree (e.g., BSc, BA)	294	27.8
	Higher degree (e.g., MSc, MA)	147	13.9
	Other	67	6.3
Religion	None	38	3.7
	Christian	154	14.8
	Muslim	728	70.1
	Hindu	53	5.1
	Sikh	48	4.6
	Any other religion	17	1.6
Indices Multiple Deprivation (IMD) Decile	1—Most deprived	12	1.1
	2	162	15.3
	3	117	11.1
	4	98	9.3
	5	109	10.3
	6	24	2.3
	7	44	4.2
	8	53	5.0
	9	14	1.3
	10—Least deprived	0	0

Ethnicity, age, and gender of Luton population is based on 2011 census statistics [33].

**Table 2 vaccines-10-00106-t002:** Sample populations intention to have a vaccination.

Vaccination Status	Subcategory	*N*	%
‘Vaccine-willing’	Already had vaccination.	322	30.43
	I definitely will accept the vaccine.	247	23.35
‘Vaccine-hesitant’	I am likely to accept the vaccine if I receive information to reassure me about its safety.	147	13.89
	I don’t feel I need the vaccine for myself, but I will accept it if it protects the wider community from COVID-19.	53	5.01
	I will only accept the vaccine if it is a requirement for travelling abroad or going to work.	96	9.07
	I definitely will not accept the vaccine under any circumstances.	53	5.01
Total		918	100

**Table 3 vaccines-10-00106-t003:** Logistic regression analysis of all explanatory variables for vaccine hesitancy.

	*B*	*SE*	Wald	df	Sig	Exp(B)	Confidence Interval	
							Lower	Upper
Age								
Age	−0.05	0.01	33.27	1	***	0.95	0.93	0.97
Gender								
Male	0.11	0.22	0.26	1	-	1.12	0.72	1.73
Female	−0.02	1.59	0.00	1	-	0.98	0.04	21.88
Other			8.68	8	-			
Ethnicity								
Indian			8.68	8	-			
Pakistani	3.46	1.93	3.12	1	-	31.67	0.72	1398.43
Bangladeshi	3.47	1.94	3.20	1	-	32.19	0.72	1441.64
Black African	3.75	1.93	3.77	1	*	42.51	0.97	1869.55
Black Caribbean	4.72	2.01	5.55	1	**	112.56	2.21	5726.78
Mixed: white and black Caribbean	4.68	2.11	4.89	1	*	107.19	1.70	6757.03
Mixed: white and Asian	−16.12	27,400.96	0.00	1	-	0.00	0.00	
Other	3.29	2.08	2.50	1	-	26.80	0.45	1581.04
Religion								
No religion			2.74	5	-			
Christian	−1.18	1.69	0.48	1	-	0.31	0.01	8.54
Hindu	−0.81	1.71	0.23	1	-	0.44	0.02	12.69
Muslim	1.40	2.30	0.37	1	-	4.07	0.05	369.13
Sikh	−0.28	1.62	0.03	1	-	0.76	0.03	18.08
Any other religion	0.59	2.42	0.06	1	-	1.80	0.02	207.27
Highest level of qualification								
No formal qualifications			2.13	5	-			
GCSE or equivalent	0.19	0.52	0.13	1	-	1.21	0.44	3.34
A-Level or equivalent	−0.30	0.50	0.36	1	-	0.74	0.28	1.96
First Degree (e.g., BSc, BA)	−0.09	0.48	0.03	1	-	0.92	0.36	2.33
Higher degree (e.g., MSc, MA)	−0.15	0.46	0.11	1	-	0.86	0.35	2.13
Other	0.18	0.52	0.12	1	-	1.19	0.43	3.28
Deprivation								
Index Multiple Deprivation (IMD) Score	0.01	0.01	0.29	1	-	1.01	0.99	1.03
Knowledge								
Levels of knowledge (symptoms)	−0.06	0.03	6.13	1	**	0.94	0.89	0.99
Levels of knowledge (transmission)	−0.47	0.16	8.36	1	***	0.63	0.45	0.86
Validating information sources								
I access information objectively to ensure information is balanced, reasonable, and unbiased.	−0.59	0.19	9.71	1	***	0.56	0.38	0.80
I ensure that information I access is from a trusted reputable well-known source.	0.24	0.15	2.49	1	-	1.27	0.94	1.72
I compare information I access with other reliable sources to ensure it is accurate.	0.08	0.17	0.21	1	-	1.08	0.78	1.50
Constant	0.95	2.47	0.15	1	-	2.60		

*B* = Standardised Beta coefficients, *SE* = Standard Error, Exp(B) exponentiated coefficient, *p* < 0.001 ***, *p* < 0.01 **, *p* < 0.05 *.

## Data Availability

The data presented in this study are openly available in The Open Science Framework. Available online: https://osf.io/5qyab/ (accessed on 8 December 2021) doi:10.17605/OSF.IO/5QYAB.

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
