# Peer review of "Vaccination against COVID-19: Factors That Influence Vaccine Hesitancy among an Ethnically Diverse Community in the UK"

_vaccines, 2022, doi:10.3390/vaccines10010106_

Round 1
Reviewer 1 Report
The manuscript provides an analysis of the findings from populations of a city in United Kingdom regarding the knowledge, understanding, certain beliefs and COVID-19 hesitancy. The data analyzed in this manuscript is based on 1058 completed surveys between January and March of 2021.
Major concerns:
- The manuscript provides in reality no new information. All of the reasons determined in this manuscript to be the factors towards vaccine hesitancy is now known. It is surprising that the authors look this long to complete the analysis of their data and put that together. So much has changed in the last 9 months after the authors completed collecting the data analyzed here regarding the understanding of COVID-19, possible of mode of transmission of the virus that is cause of this disease, the disease symptoms, availability of a variety of different vaccines worldwide. This could have been more useful if it was published immediately after the data was complied.
- It seems that the authors used many different approached to reach individual survey responders. However, it appears that majority of the respondents came from Pakistan and Bangladesh. Does this mean that the approached used to target the survey responders was biased towards specific communities?
- One major aspect missing in the analysis of the data is the religious beliefs of the survey responders. It appears that the majority of the survey responders are Muslims and if their religious beliefs are anyway also contributing to vaccine hesitancy.
Minor Concerns: The manuscript lacks clarity in many places. It must provide better clarity by restating some of the sentences. For example:
- Line 47: COVID-19 is a disease. Infection is caused by SARS-CoV2.
- Line 49: should be “effective respectively”
- Line 58: sufficient protection
- Line 82-87: Extremely unclear
- Line 90: be specific with the sociodemographics (what age? What level of education? What income bracket?)
- Line 102: reflective “of” their
- Line 107: Use better language than ‘This study addresses this….
- Line 116-121: poorly worded
- Line 127: explain deprivation
- Line 127-129: poorly worded
- Line 139: on the questionnaire and in the campaigns what about adjustments for non-English/ non-Urdu speakers?
- Line 145: What about individuals who can’t write in English, how were they expected to complete questionnaires?
- Line 187-192: Not sure if these are the best options for a survey?
- Line 207: classifying those who select “I will not take the vaccine under any circumstances” as vaccine “hesitant” is misleading
- Line 212: why are the p-values not consistent
Additionally:
- Why not utilize a Likert scale, yes/no questionnaires are limiting
- There was also no option for ‘I do not know’ on the questionnaire
- For question 2.2.4, there is no option for individuals who may not have taken the vaccine due to health issues
- The explanation for the qualitative analysis is unclear and confusing
- There are only 44 responses for the qualitative portion, not generalizable
- For question 3.2.1. attitudes and beliefs should be in separate categories
- Graphs a/b lacked titles and were unimpressive, the information was already written and the graphs themselves added nothing
Author Response
We would firstly like to thank you for your useful feedback and are pleased to confirm we have addressed all comments as set out below:
R1 C1: The manuscript provides in reality no new information. All of the reasons determined in this manuscript to be the factors towards vaccine hesitancy is now known. It is surprising that the authors took this long to complete the analysis of their data and put that together. So much has changed in the last 9 months after the authors completed collecting the data analyzed here regarding the understanding of COVID-19, the mode of transmission of the virus that is the cause of this disease, the disease symptoms, availability of a variety of different vaccines worldwide. This could have been more useful if it was published immediately after the data was compiled.
This was a co-commissioned piece of work with Luton public health, so is informing their current vaccine/booster rollout etc, hence our priority has been to focus on applying our findings in practice rather than publishing them asap. Overall, our work will be useful as the UK seeks to increase uptake of non-vaccinated groups and increase uptake of boosters in hard to engage communities.
R1 C1: It seems that the authors used many different approaches to reach individual survey responders. However, it appears that the majority of the respondents came from Pakistan and Bangladesh. Does this mean that the approached used to target the survey responders were biased towards specific communities?
The survey was circulated for the attention of the Indian, Bangladeshi, Indian, black African and black Caribbean communities. The approach to circulating the survey was the same for all our selected communities. The responses are representative of the size of these communities in Luton (See Table 1).
R1 C3: One major aspect missing in the analysis of the data is the religious beliefs of the survey responders. It appears that the majority of the survey responders are Muslims and if their religious beliefs are anyway also contributing to vaccine hesitancy.
Thank you for this useful observation. We can now confirm that religion has now been entered as an explanatory variable in the analysis.
R1 C4: The manuscript lacks clarity in many places. It must provide better clarity by restating some of the sentences. For example:
Line 47: COVID-19 is a disease. Infection is caused by SARS-CoV2.
This has now been changed.
Line 49: should be “effective respectively”
This has now been changed.
Line 58: sufficient protection
This has now been changed.
Line 82-87: Extremely unclear
This sentence has been changed to make it clearer.
Line 90: be specific with the sociodemographic (what age? What level of education? What income bracket?)
We have been more specific with the socio-demographic factors this section now reads: ‘Sociodemographic factors include being female[1], being younger (aged 16-24 years) [1-3], lower educational attainment (no formal qualifications)[1, 4, 5] and lower-income (<£16,000)[2, 5, 6] have all been linked to vaccine hesitancy’.
Line 102: reflective “of” their
This has now been changed.
Line 107: Use better language than ‘This study addresses this….
This has now been deleted as the aim is already discussed in the ‘Material and Methods’ section.
Line 116-121: poorly worded
This has been rephrased and made more concise. This now reads ‘Luton is an ethnically diverse town with a population of around 216,000 people situated in the East of England. Luton is only one of three towns in the UK to host a population whereby less than 50% of the population identify as white British[7]. Luton also experiences high levels of relative deprivation (measured by the Index Multiple Deprivation[8]) compared to other parts of the UK[9], where large sections of the ethnically diverse community reside’.
Line 127: explain deprivation
This was measured using the Index of Multiple Deprivation (IMD), this has been made clearer with a supporting citation provided.
Line 127-129: poorly worded
This has now been rephrased.
Line 139: on the questionnaire and in the campaigns what about adjustments for non-English/ non-Urdu speakers?
We did not adjust the survey i.e., translate into community languages because many people do not read their vernacular. Instead, our TLC Community Researchers interviewer administered the survey for those people who were unable to read the questionnaire. This is how we were able to reach hard to reach individuals. We have now added into the methods section ‘TLC Community Researchers interviewer administered the survey for those people who were unable to read the questionnaire. A total of 1,058 individuals completed the survey’.
Line 145: What about individuals who can’t write in English, how were they expected to complete questionnaires?
TLC community researchers administered the questionnaires so were able to support those who could not write in English. Family/friends also supported those unable to complete if not supported by a TLC community researcher.
Line 187-192: Not sure if these are the best options for a survey?
These categories were used as they were developed by the social marketing company (NSMC) who had previously engaged with the Luton population and developed these to segment the population.
Line 207: classifying those who select “I will not take the vaccine under any circumstances” as vaccine “hesitant” is misleading
We have now excluded those who selected ‘I will not take the vaccine under any circumstances’ from the analysis.
Line 212: why are the p-values not consistent
These have now been made more consistent.
R1 C5: Why not utilize a Likert scale, yes/no questionnaires are limiting
The authors acknowledge that more use of likert scale responses would have been less restrictive in terms of analysis; however, all questionnaires were developed using previously published surveys all of which have been acknowledged within our methods section.
R1 C6: There was also no option for ‘I do not know’ on the questionnaire
We did provide an ‘other’ option for the socio-demographic characteristics. We did also include an ‘I do not know’ option for the knowledge questions and this has now been added.
R1 C7: For question 2.2.4, there is no option for individuals who may not have taken the vaccine due to health issues
We acknowledge that this was not an available option for participants; however, this should have been picked up in the open text fields to which they would have been directed. We can also confirm that no participants in our sample cited medical exemptions as a reason for not having the vaccine.
R1 C8: The explanation for the qualitative analysis is unclear and confusing
This explanation has been rewritten to improve clarity both in the methods and in the abstract.
R1 C9: There are only 44 responses for the qualitative portion, not generalizable
As there was only a small number of people invited based on the uptake (5% of the total sample) and therefore we were by default going to get a relatively small number of individuals answering Q1 and Q2. Nonetheless, the aim of the qualitative element of the sample was not to obtain generalisable results, but an insight into the reasons that inhabitants of Luton may have for not wanting to have a vaccination, articulated in their own terms (we have inserted ‘may’ in the Discussion to stress this). Hence, we do not attempt to make statistical statements regarding how frequent given views are, but to reveal some of the views held by individuals which they express as reasons for not getting vaccinated. Having identified them, future researchers can design further surveys to establish how common they are.
R1 C10: For question 3.2.1. attitudes and beliefs should be in separate categories
This category has been renamed to ‘beliefs’.
R1 C11: Graphs a/b lacked titles and were unimpressive, the information was already written and the graphs themselves added nothing
These graphs have now been deleted.
Reviewer 2 Report
The paper investigates perceptions of minority groups, quantitatively and qualitatively, in a UK city towards vaccines, as this group tends to be vaccine hesitant. The survey recruitment, questionnaire, and analysis is of strong quality. The supplementary qualitative study adds depth to the survey findings.
Apart from fixing the writing errors, my only suggestion is to add depth to the analysis section by investigating what makes this diverse subgroup hesitant to vaccination. Are there specific cultural factors and historical experiences that makes them vulnerable to hesitancy. Do the strong family and friendship ties make them susceptible to fake news or influence of one elderly skeptic? Without such insights, the study fails to do justice to understanding this subgroup.
Author Response
We would firstly like to thank you for your useful feedback and are pleased to confirm we have addressed all comments as set out below:
R2 C1: Apart from fixing the writing errors, my only suggestion is to add depth to the analysis section by investigating what makes this diverse subgroup hesitant to vaccination. Are there specific cultural factors and historical experiences that make them vulnerable to hesitancy? Do the strong family and friendship ties make them susceptible to fake news or the influence of one elderly sceptic? Without such insights, the study fails to do justice to understanding this subgroup.
Thank you for your comments. In response, we have within the statistical analysis added religion and assessing trustworthiness of sources to the regression model which has added some further insight into predictors of vaccine hesitancy. We have also tried to consider some of these factors within the discussion, particularly when discussing the qualitative findings which uncovered trust in the government as an important reason for not wanting to have a vaccine. Nonetheless, the authors agree that there may be particular cultural and historical experiences that have not been assessed which may make them feel more vulnerable to vaccine hesitance and as such we have suggested within the discussion that future research should incorporate this subject matter for more detailed exploration.
Reviewer 3 Report
suggestions by line:
l.47: Brand names could be avoided [Pfizer & Astra] and replaced by biological names
l.75: Inequity in availability or access to vaccination is not considered
Lines 114 starting from “The overall aim of this project” to l. 121 …”take up in these communities” have to be placed after line 110 and before the “Material and Methods” chapter as they are referring to the aims of the study.
l. 130: the inclusion methodology is mainly based with the prerequisite that the participants are computer and internet literates. This is excluding possibly the most deprived, lower education and oldest part of the population which could possibly have higher rates of vaccination hesitancy compared with the rest of the participants. As this can not be changed I suggest to include a relevant phrase in the limitations part of the paper.
l.162: Table 1 could better fit as the first part of the “Results” chapter.
l.162: in Table 1 it would be helpful to show a comparison of the distribution of the socio demographic characteristics between the sample and the target population [percentages of the different ethnic minorities].
l.248-49 and 263-264 are both referring to the association of age and willingness to have a vaccine. No need to repeat.
The regression ORs are sufficient but are wrong in their direction. . If higher age is linked with increased willingness in your model, then the OR would be expected to be more than 1 [including the CIs].
The same mistake is written in the title of the Table 3. All the results are probably resulting from “a regression analysis of all explanatory variables to vaccine hesitancy “ instead of “accept vaccination”. E.g. OR = 0,95 means that age is a protective factor against hesitancy, or being Black Caribbean is a risk factor for hesitancy.
There is a need to include the Index of Multiple Deprivation scores in your analysis. Especially when studying ethnic minorities, you have to control for socioeconomic characteristics.
l.271
The following phrase is not clear, need to be recomposed.
l. 271: “differences were found for participants having a self-disclosed chronic health condition, education level and deprivation made no significant contribution to explaining vaccine intent”…
Author Response
We would firstly like to thank you for your useful feedback and are pleased to confirm we have addressed all comments as set out below:
R3 C1: l.47: Brand names could be avoided [Pfizer & Astra] and replaced by biological names
This has now been replaced with biological names and reads ‘Vaccines BNT162b2 and ChAdOx1-S, are the most commonly used in the UK, shown to be 90%[10] and 82%[11] effective respectively against severe disease to the original variant’.
R3 C2: l.75: Inequity in availability or access to vaccination is not considered
Thank you for highlighting this important point, we can confirm that this has now been included within the introduction.
R3 C3: Lines 114 starting from “The overall aim of this project” to l. 121 …”take up in these communities” have to be placed after line 110 and before the “Material and Methods” chapter as they are referring to the aims of the study.
This section has been deleted as the aim is already discussed in the ‘Material and Methods’ section.
R3 C4: l. 130: the inclusion methodology is mainly based on the prerequisite that the participants are computer and internet literates. This is excluding possibly the most deprived, lower education and oldest part of the population which could possibly have higher rates of vaccination hesitancy compared with the rest of the participants. As this cannot be changed I suggest including a relevant phrase in the limitations part of the paper.
It is also important to highlight that as well as online surveys we distributed printed surveys which were disseminated through a wide range of networks and community groups as outlined in the Methods section. This, therefore, meant that those who did not have a computer or are internet literate could still take part. The majority of participants who took part in the survey did so via the paper-based version. We felt it is important to reflect on this, so we have now added the completion rates via each method in the results section ‘Most participants completed the paper-based survey (n=761; 72.7%) with the remaining 27.3% (n=286) who completed the survey online’. We also used COVID-19 Champions and TLC Community Researchers to help recruit and administer the survey for those people who were unable to read the questionnaire. This has now been acknowledged more clearly within the methods section.
R3 C5: l.162: Table 1 could better fit as the first part of the “Results” chapter.
This has now been moved to the Results section.
R3 C6: l.162: in Table 1 it would be helpful to show a comparison of the distribution of the socio-demographic characteristics between the sample and the target population [percentages of the different ethnic minorities].
We have now added census socio-demographic distributions for Luton (where relevant) in Table 1 for comparison.
R3 C7: l.248-49 and 263-264 are both referring to the association of age and willingness to have a vaccine. No need to repeat.
Lines 248-49 are referring to a chi-square analysis, whereas L263-64 is referring to the logistic regression model. This has now been made clearer.
R3 C8: The regression ORs are sufficient but are wrong in their direction. If higher age is linked with increased willingness in your model, then the OR would be expected to be more than 1 [including the CIs]. The same mistake is written in the title of Table 3. All the results are probably resulting from “a regression analysis of all explanatory variables to vaccine hesitancy “ instead of “accept vaccination”. E.g. OR = 0,95 means that age is a protective factor against hesitancy, or being the Black Caribbean is a risk factor for hesitancy.
Thank you for highlighting this. This was an error and has now been amended. The title of Table 3 has also been changed to ‘Table 3: Logistic regression analysis of all explanatory variables for COVID-19 vaccine hesitancy’.
R3 C9: There is a need to include the Index of Multiple Deprivation scores in your analysis. Especially when studying ethnic minorities, you have to control for socioeconomic characteristics.
IMD score has now been added to the analysis.
R3 C10: l.271 The following phrase is not clear, need to be recomposed.
This has now been rephrased.
R3 C11: l. 271: “differences were found for participants having a self-disclosed chronic health condition, education level and deprivation made no significant contribution to explaining vaccine intent”…
This has been changed to ‘Self-disclosed chronic health condition, education level and deprivation made no significant contribution to explaining vaccine hesitancy.
Round 2
Reviewer 1 Report
The authors have addressed the concerns I had as best as they could.